# Analysis of Different Strains Fermented Douchi by GC×GC-TOFMS and UPLC–Q-TOFMS Omics Analysis

**DOI:** 10.3390/foods13213521

**Published:** 2024-11-04

**Authors:** Liqiang Sui, Sugui Wang, Xin Wang, Lingling Su, Huilong Xu, Wei Xu, Lixia Chen, Hua Li

**Affiliations:** 1College of Pharmacy, Fujian University of Traditional Chinese Medicine, Fuzhou 350122, China; suiliqiang007@163.com (L.S.);; 2Institute of Structural Pharmacology & TCM Chemical Biology, Fujian University of Traditional Chinese Medicine, Fuzhou 350122, China; syzyclx@163.com; 3Fujian Key Laboratory of Chinese Materia Medica, Fuzhou 350122, China; 4Wuya College of Innovation, Shenyang Pharmaceutical University, Shenyang 110016, China; 5Key Laboratory of Structure-Based Drug Design & Discovery, Ministry of Education, Shenyang 110016, China

**Keywords:** Douchi, *Aspergillus niger*, *Rhizopus arrhizus*, *Bacillus circulans*, GC×GC-TOFMS, UPLC–Q-TOFMS, flavor compounds, non-volatile components

## Abstract

Douchi is a kind of soybean-fermented food in China. To explore the common and differential compounds in different Douchi, Douchi was fermented by *Aspergillus niger*, *Rhizopus arrhizus*, and *Bacillus circulans*, respectively, and co-fermented by the three strains in this study. The common and characteristic flavor compounds and common and characteristic non-volatile components of different strains of fermented Douchi were explored through GC×GC-TOFMS and UPLC–Q-TOFMS omics analysis. The result suggested that Pyrazines, ketones, and alkenes such as tetramethyl-pyrazine, 2,5-dimethyl pyrazine, furaneol, 2,3-butanedione, gamma-terpinene might contribute to the basic flavor of the Douchi fermented by *A. niger*, *R. arrhizus*, and *B. circulans*. Peptides, amines, and flavonoids, such as N–acetylhistamine, 7,3′,4′–trihydroxyflavone, (3S,8As)-3-isobutylhexahydropyrrolo[1,2-a]pyrazine-1,4-dione might contribute to the basic function of the above three Douchi. The common metabolic pathways involved in the fermentation were isoflavonoid biosynthesis, flavonoid biosynthesis, etc. Ketones and esters such as 2,3-dihydro-3,5-dihydroxy-6-methyl-4H-pyran-4-one, 3-octanone, 5-methylfurfural and nonanal contributed to the unique flavor, while betaine, oleanolic acid, saikosaponin D and leucine might contribute to the unique function of *A. niger* fermented Douchi. Alkenes, pyrazine, and ketones such as α-terpinene, ethyl-pyrazine, dihydro-3-methyl-2(3H)-furanone, and linalool might contribute to unique flavor, while cordycepin, 2-Phenylacetamide might contributed to the unique function of *R. arrhizus* fermented Douchi. The unique flavor of *B. circulans* fermented Douchi might derived from ketones and esters such as 3-acetyl-2-butanone, 2-tridecanone, propionic acid-2-phenylethyl ester, while vitexin, astragalin, and phenethylamine might contribute to the unique function. Compared with single-strain fermented Douchi, the flavor substances and non-volatile components in multi-strain fermented Douchi were more abundant, such as hexadecanoic acid methyl ester, benzeneacetic acid ethyl ester, 9,12-octadecadienoic acid ethyl ester, nuciferine, and erucamide. It was speculated that there were common and differential substances in Douchi fermented by *Aspergillus niger*, *Rhizopus arrhizus*, and *Bacillus circulans*, which might contribute to the basic and unique flavor and function. Compared with single-strain fermented Douchi, the flavor substances and metabolites in multi-strain fermented Douchi were more abundant. This study provided a reference for the research of flavor and functional substances of Douchi.

## 1. Introduction

Douchi is a kind of soybean (seeds of *Glycine max* (L.) Meorr.) fermented food in China similar to Natto in Japan and Tempeh in Indonesia. There are a variety of functional active substances in Douchi, such as proteins, amino acids, polysaccharides, saponins, flavonoids, etc. Some functional substances are generated from fermentation transformation under the action of microorganisms. For example, the proteins are decomposed into amino acids, polysaccharides, and other macromolecules are enzymatically hydrolyzed into small molecules such as oligosaccharides or monosaccharides [1,2]. The functions of Douchi include anti-oxidation, lowering blood pressure, lowering blood lipids, anti-osteoporosis, anti-thrombosis, and regulating intestinal health [3,4,5].

The recent research on fermentation strains in Douchi has attracted some attention. It has been found that *Mucor*, *Aspergillus*, *Rhizopus*, *Bacillus subtilis*, and other microorganisms were involved in the fermentation of Douchi [6,7,8]. The single-strain fermentation processes of Douchi have been reported, such as *wickhamomyces anomalus* fermented Douchi, *Bacillus amyloliquefaciens* fermented Douchi [9,10], etc. One study reported that there were obvious differences in metabolites between fungal fermented Douchi and bacterial fermented Douchi, the formation of branched aliphatic alcohols and esters increased in fungal fermented Douchi, while the content of fatty acid volatiles was higher in bacterial fermented Douchi [11]. However, the above study was not systematic and in-depth, and there was no systematic summary of the common rules and characteristic substances of different strains of fermentation. There were few reports on the difference between single-strain fermented Douchi and multi-strain fermented Douchi. The difference between various strains of fermented Douchi and the difference between single-strain and multi-strain fermentation are worth further studying.

On the basis of previous experiment and literature analysis, we speculate that the functional substances of different strains fermented Douchi will be different, and there should be some common basic transformation in the fermentation. Co-fermentation of multiple strains may produce more functional compounds under the action of multiple microorganisms.

Gas chromatography–mass spectrometry (GC-MS) and liquid chromatograph mass spectrometer (LC-MS) are common methods to analyze the differences between flavor and non-volatile components [12,13]. This study chose comprehensive two-dimensional gas chromatography–time of flight mass spectrometry (GC×GC-TOFMS) and Ultra Performance Liquid Chromatography Quadrupole-Time of Flight Mass (UPLC–Q-TOFMS) omics analysis as analysis methods. GC×GC-MS has the highest sensitivity and is suitable for the analysis of complex samples, which are difficult to be separated by one chromatographic column. Better separation of sample components can obtain more pure analysis target chromatographic peaks, eliminate the interference of homologs or substrates, and significantly improve the qualitative and quantitative accuracy. UPLC–Q-TOFMS omics analysis mainly relies on high-resolution mass spectrometers, which can perform unbiased, large-scale, and systematic detection of various metabolites in the sample and reflect the metabolic level disturbance in the organism to the greatest extent [14,15].

Our research group found that the main strains in the fermentation of Douchi were *Aspergillus niger*, *Rhizopus arrhizus*, and *Bacillus circulans* in previous experiments. Therefore, Douchi was fermented by the above three strains, respectively, and co-fermented by the three strains. The common and characteristic flavor compounds and non-volatile components of different strains of fermented Douchi were explored through GC×GC-TOFMS combined with UPLC–Q-TOFMS omics analysis. The advantages of multi-strain fermented Douchi were further explored by the above methods. The study may provide a reference for the research of flavor and functional substances of Douchi and the development of new soybean-fermented functional food.

## 2. Materials and Methods

### 2.1. Materials and Microorganisms

Black soybean (commercially available) was identified by Che Surong, associate professor at Fujian University of Traditional Chinese Medicine, as seeds of *Glycine max* (L.) Meorr. *B. circulans* (BWCC30302), *A. niger* (BWCC672840), and *R. arrhizus* (BWCC67202) were obtained from Henan Standard Substance Research and Development Center. LB medium (211206, Hangzhou Binhe Microbial Reagent Company, Hangzhou, China), PDA medium (BS112, Hangzhou Baisi Biotechnology Company, Hangzhou, China).

### 2.2. Solid-State Fermentation of Douchi

*A. niger* fermented Douchi: 5% *A. niger* was inoculated into black beans, fermented at 30 °C for 8 days. Then Douchi was dried at 50 °C after washing with water.

*R. arrhizus* fermented Douchi: 5% *R. arrhizus* was inoculated into black beans, fermented at 30 °C for 8 days. Then Douchi was dried at 50 °C after washing with water.

*B. circulans* fermented Douchi: 5% *B. circulans* was inoculated into black beans fermented at 30 °C for 8 days. Then Douchi was dried at 50 °C after washing with water.

Three strains of co-fermented Douchi: 5% mixed strains (*B. circulans*:*A. niger*:*R. arrhizus* = 1:2:1) were inoculated into black beans, fermented at 30 °C for 8 days. Then Douchi was dried at 50 °C after washing with water.

### 2.3. Volatile Components Analysis

Five samples of each fermented Douchi were randomly selected from 5 batches as analytical samples. The volatile components were determined by SPME-GC×GC–TOFMS according to the description of Shen et al. [16]. Volatile components were extracted by a 50/30 μm divinylbenzene/carboxy/polydimethylsiloxane (DVB/CAR/PDMS) fiber at 60 °C for 30 min. The solid phase microextraction (SPME) fiber was desorbed at the GC injector with the condition of 245 °C for 5 min and analyzed by GC×GC–TOFMS.

The GC×GC–TOFMS system (Agilent 7890B GC–TOFMS spectrometer, Agilent Technology Co., Ltd., Santa Clara, CA, USA, LECO Pegsus BT 4D mass spectrometer, LECO Laboratory Equipment Corporaton, Saint Joseph, MO, USA) equipped with a DB-WAX capillary column (30 m × 0.25 mm, 0.25 μm film thickness) and a DB-17SilMS capillary column (2 m × 0.1 mm, 0.1 μm film thickness). The flow rate of the carrier gas helium was 1.0 mL/min. The injector temperature was 245 °C, and the split ratio was 5:1. The electron ionization source temperature was 220 °C. The GC oven temperature was maintained at 40 °C for 1 min and then increased at a rate of 4 °C/min to 200 °C. Finally, the temperature was increased to 240 °C at a rate of 10 °C/min and maintained for 5 min. The ionization energy was 70 eV. The mass spectrometer was operated in the full-scan-monitoring mode with a mass range of 30–450 *m*/*z* [17,18].

The volatile components were identified based on the National Institute of Standards and Technology (NIST 23) Mass Spectral Library, and the relative percentage content analysis was performed based on the peak areas.

### 2.4. Non-Volatile Components Analysis

Five samples of each fermented Douchi were randomly selected from 5 batches as analytical samples. A 1 g sample powder was added to 20 mL methanol for ultrasonic extraction for 30 min, centrifuged, and the supernatant was injected as extracting solution.

UPLC–Q-TOFMS system (Agilent 1290 Infinity UPLC, Agilent Technology Co., Ltd., Santa Clara, CA, USA, AB Triple TOF 6600 mass spectrometer, AB Sciex Analytical Instrument Trading Co., Ltd., Boston, MA, USA) equipped with a Waters, ACQUITY UPLC BEH C-18 column (100 mm × 2.1 mm, 1.7 μm). The flow rate was 0.4 mL/min. The column temperature was 40 °C. The injection volume was 2 μL. The mobile phase was composed of water (containing 25 mM ammonium acetate and 0.5% formic acid) and methanol. The elution method was gradient elution.

The ESI lon Source Gas1 and Gas2 nebulizer pressure was 60 Psi. The curtain gas nebulizer pressure was 30 Psi. The source temperature was 600 °C. The ion spray voltage floating (ISVF) was ±5500 V (positive and negative modes). TOFMS scan *m*/*z* range was 60–1000 Da, product ion scan *m*/*z* range was 25–1000 Da. Secondary mass spectrometry was obtained by information-dependent acquisition (IDA).

The structures of the non-volatile components were identified by matching the molecular weight, secondary fragmentation spectrum, retention time, and other information of the components in the in-house database (Shanghai Applied Protein Technology, Shanghai, China). MS-DIAL ver.4.60 software was used for peak alignment, retention time correction, and extraction peak area. The data extracted from MS-DIAL ver.4.60 were first subjected to metabolite structure identification and data preprocessing, then the quality of the experimental data was evaluated, and finally, the data analysis was performed by in-house database (Shanghai Applied Protein Technology).

## 3. Results and Discussion

### 3.1. Changes in Volatile Components in Different Strains Fermented Douchi

There were a total of 1151 volatile substances identified by GC×GC–TOFMS. The total ion current chromatogram for quality control was shown in Figure 1B, and the PCA analysis of different groups of Douchi was shown in Figure 1A. The main substances included alkanes, nitrogen compounds, esters, alcohols, and ketones (Figure 1C). There were 173 different compounds between *A. niger*-fermented Douchi and *R. arrhizus*-fermented Douchi (Figure 2C). There were 165 different compounds between *A. niger*-fermented Douchi and *B. circulans*-fermented Douchi (Figure 2B). There were 151 different compounds between *R. arrhizus* fermented Douchi and *B. circulans* fermented Douchi (Figure 2A).

The common changes in the above three strains fermented Douchi were 58 volatile substances upregulated and 33 volatile substances downregulated after fermentation. Among them, the downregulated volatile substances mainly included alcohols and alkanes, and the upregulated volatile substances mainly included pyrazine (24%), ketones (26%), and alkenes, such as tetramethyl-pyrazine, 2,5-dimethyl pyrazine, furaneol, 2,3-butanedione, gamma-terpinene, etc.

Further analysis showed that there were 43 characteristic volatile substances in *A. niger* fermented Douchi, among which ketones (40%), esters, and aldehydes were main flavor compounds such as 3-octanone, 2,3-dihydro-3,5-dihydroxy-6-methyl-4H-pyran-4-one, propanoic acid pentyl ester, nonanal, etc. There were 13 characteristic volatile substances in fermented Douchi fermented by *R. arrhizus*, among which alkenes, pyrazine, and ketones were the main flavor compounds, such as α-terpinene, ethyl-pyrazine, dihydro-3-methyl-2(3H)-furanone, etc. There were 15 characteristic volatile substances in *B. circulans* fermented Douchi, among which ketones and ester were main flavor compounds such as 3-acetyl-2-butanone, 2-tridecanone, propionic acid-2-phenylethyl ester, etc.

### 3.2. Changes in Volatile Components in Single-Strain and Multi-Strain Fermented Douchi

There were 180 different compounds between multi-strain fermented Douchi and *A. niger* fermented Douchi (Figure 3A), 158 different compounds between multi-strain fermented Douchi and *R. arrhizus* fermented Douchi (Figure 3C), 153 different compounds between multi-strain fermented Douchi and *B. circulans* fermented Douchi identified by GC×GC–TOFMS (Figure 3B).

The common changes in multi-strain fermented Douchi and three pure-strain fermented Douchi were 54 volatile substances upregulated and 23 volatile substances downregulated after fermentation, which was similar to the common changes in the above three pure-strain fermented Douchi.

Further analysis showed that there were 18 characteristic volatile substances in multi-strain fermented Douchi among which esters (28%) and ketones were main flavor compounds such as hexadecanoic acid methyl ester, benzeneacetic acid ethyl ester and 9,12-octadecadienoic acid ethyl ester, (E)-9-octadecenoic acid ethyl ester, etc.

### 3.3. Changes in Non-Volatile Components in Different Strains Fermented Douchi

There were a total of 1568 non-volatile components identified by UPLC–Q-TOFMS; 844 components were identified in positive ion mode, and 724 components were identified in negative ion mode. The total ion current chromatograms for quality control were shown in Figure 4A,B, and the PCA analysis of different groups of Douchi was shown in Figure 4C. The main non-volatile components included prenol lipids, flavonoids, carboxylic acids and derivatives, benzene and substituted derivatives, isoflavonoids, and glycerophospholipids (Figure 4D). There were 145 different non-volatile components between *A. niger*-fermented Douchi and *R. arrhizus*-fermented Douchi (Figure 5B). There were 124 different non-volatile components between *A. niger*-fermented Douchi and *B. circulans*-fermented Douchi (Figure 5C). There were 102 different non-volatile components between *R. arrhizus*-fermented Douchi and *B. circulans*-fermented Douchi (Figure 5A).

The common changes in the above three strains fermented Douchi were 14 non-volatile components upregulated and 36 non-volatile components downregulated after fermentation. Among them, the downregulated non-volatile components mainly included glycosides and sugars such as thermopsoside, petunidin 3-galactoside, 6″-O-malonylgenistin, etc. The upregulated non-volatile components mainly included peptides, amines, and flavonoids, such as (3S,8As)-3-isobutylhexahydropyrrolo[1,2-a]pyrazine-1,4-dione, N–acetylhistamine, 7,3′,4′–trihydroxyflavone, etc. The result suggested speculation during the fermentation process: under the action of three microorganisms, sugars were metabolized, glycosidic bonds were broken to provide nutrients, and new functional substances were generated.

Further analysis showed that there were 36 characteristic non-volatile components in *A. niger*-fermented Douchi, such as betaine, oleanolic acid, saikosaponin D, leucine, etc. There were eight characteristic non-volatile components in Douchi fermented by *R. arrhizus*, such as cordycepin, 2-Phenylacetamide, etc. There were 24 characteristic non-volatile components in *B. circulans* fermented Douchi, among which such as vitexin, astragalin, phenethylamine, etc.

### 3.4. Changes in Non-Volatile Components in Single-Strain and Multi-Strain Fermented Douchi

There were 147 different non-volatile components between multi-strain fermented Douchi and *A. niger* fermented Douchi (Figure 6A), 135 different non-volatile components between multi-strain fermented Douchi and *R. arrhizus* fermented Douchi (Figure 6C), 98 different non-volatile components between multi-strain fermented Douchi and *B. circulans* (Figure 6B) fermented Douchi identified by UPLC–Q-TOFMS.

Further analysis showed that there were seven characteristic non-volatile components in multi-strain fermented Douchi such as nuciferine, erucamide, etc.

### 3.5. The Enriched KEGG Pathways Analysis of the Three Microorganisms Fermented Douchi and Multi-Strain Fermented Douchi

Through KEGG pathway annotation and enrichment of metabolites, the main metabolic pathways with higher abundance were emphatically analyzed. The common metabolic pathways involved in the fermentation of Douchi by the above three microorganisms were isoflavonoid biosynthesis, flavonoid biosynthesis, flavone and flavonol biosynthesis, and longevity regulating pathway (Figure 7A–D).

### 3.6. Discussion

In this study, the common components, unique flavor substances, and metabolites of Douchi fermented by different strains were screened through GC×GC-TOFMS and UPLC–Q-TOFMS omics analysis. Some flavor substances and metabolites were consistent with the results of the study [3,11], but a considerable number of compounds were different due to different strains. The result proved that strains played an essential role in the flavor and metabolites of Douchi. In addition, this study focused on the common components of Douchi fermented by different strains and the change in multi-strain fermented Douchi less involved in previous studies.

Among the volatile components screened in this study, the aroma of 2,5-dimethyl pyrazine is described as a fried peanut fragrance, chocolate, and creamy smell [19]. The aroma of 2-ethyl-3,5- dimethyl pyrazine is described as cocoa and baking. The aroma of 2,3-Butanedione is described as caramel and cheese aromas [20]. The aroma of γ-terpinene is described as woody and lemony aroma [21]. It was speculated that the above flavor compounds contributed to the basic flavor of the three strains of fermented Douchi.

The aroma of 2,3-dihydro-3,5-dihydroxy-6-methyl-4H-pyran-4-one is characterized by fragrance, which plays a head aroma role in chili oil [22]. The flavor characteristics of 3-octanone were fruity and nutty. The flavor characteristic of 5-methylfurfural is baking aroma [23]. The flavor characteristics of nonanal are citrus and cucumber aroma [20]. The flavor characteristic of *γ*-nonalactone is milky. The flavor characteristics of *γ*-octalactone are coconut and nutty flavor [24]. It was speculated that the above flavor compounds contributed to the unique flavor of *A. niger* fermented Douchi.

The flavor characteristics of *α*-terpinene are woody and lemony [25]. The flavor characteristics of ethyl pyrazine are vegetables and fruits, which are related to the flavor of potato after puffing [26]. The flavor of dihydro-3-methyl-2 (3H)-furanone is stir-fried peanut and nut [27]. The aroma of linalool is described as floral, citrus, woody, and sweet [28]. It was speculated that the above flavor compounds contributed to the unique flavor of *R. arrhizus* fermented Douchi.

The odor of 3-acetyl-2-butanone is described as creamy [29]. The odor of 2-tridecanone is described as herbal and oil [30]. The odor of propionic acid-2-phenylethyl ester is described as honey and strawberry flavor [31]. It was speculated that the above flavor compounds might contribute to the unique flavor of *B. circulans* fermented Douchi.

The flavor characteristics of hexadecanoic acid methyl ester, benzeneacetic acid ethyl ester, 9,12-octadecadienoic acid ethyl ester, and (E)-9-octadecenoic acid ethyl ester were described as floral and fruity [23,32], which might contribute to the unique flavor of the three multi-strain fermented Douchi.

Among the non-volatile components screened in this study, betaine is a methyl derivative of glycine, which can regulate cell osmotic pressure, maintain body fluid balance, and protect the structural and functional integrity of cells, proteins and enzymes. In addition, betaine is also involved in methionine metabolism and promotes creatine and muscle protein synthesis [33]. Oleanolic acid is effective in anti-inflammatory, anti-oxidation, liver protection, hypoglycemic, lipid-lowering, anti-cancer, enhancing immunity, and so on [34,35]. It was speculated that the above metabolites might contribute to the unique function of *A. niger*-fermented Douchi.

Cordycepin has a variety of biological activities, such as antibacterial, antiviral, antitumor, antioxidant, anti-aging, immune regulation, and so on [36,37], which might contribute to the unique function of *R. arrhizus* fermented Douchi.

Vitexin possesses a variety of bioactive properties, including antioxidation, anti-inflammation, anti-cancer, neuron-protection, and cardio-protection [38]. Astragalin has a wide range of pharmacological activities and possesses therapeutic effects against a variety of diseases [39], covering cancers, obesity, diabetes mellitus, etc. It was speculated that the above metabolites might have contributed to the unique function of *B. circulans* fermented Douchi.

Neferine might contribute to the unique function of multi-strain fermented Douchi and has the effects of lipid-lowering, weight loss, and liver protection. It has good biological activities in regulating lipid metabolism and glucose metabolism, inhibiting tumor growth, regulating bone metabolism, and protecting nerves, which has good development and application prospects [40].

The above result suggested that there were common and differential substances in Douchi fermented by different strains, which might have contributed to the basic and unique flavor and function.

The result of the enriched KEGG pathways analysis showed that the common metabolic pathways of Douchi fermented by the three strains were isoflavonoid biosynthesis, flavonoid biosynthesis, etc. It was speculated that the above metabolic pathways involved in the metabolites contributed to the basic function of the three strains of fermented Douchi.

UPLC–Q-TOFMS omics analysis can perform unbiased, large-scale, and systematic detection of various metabolites in the sample and reflect the changes in metabolites in organisms to the greatest extent, relying on the high-resolution mass analyzer of the high-resolution mass spectrometer, which is suitable for the basic research in the early stage of the project. There are also limitations of this method, such as the lack of further identification of most metabolic analysis results, the possibility of partial deletion of metabolites, and the limited known metabolic pathways.

This study did not identify compounds against standards screened through GC×GC-TOFMS and UPLC–Q-TOFMS omics analysis due to the limitation of research funds and time. We will further identify the valuable compounds, explore the biosynthesis mechanism of the compounds, and try to develop Douchi products with different flavors in subsequent studies.

## 4. Conclusions

The flavor substances and metabolites in Douchi fermented by *Aspergillus niger*, *Rhizopus arrhizus*, and *Bacillus circulans* were significantly different, while there were common changes in the fermentation. The different flavor substances and metabolites might contribute to the unique flavor and function of Douchi fermented by different strains. The common changes might contribute to the basic flavor and function of Douchi fermented by the three strains to a certain degree. Compared with single-strain fermented Douchi, the flavor substances and metabolites in multi-strain fermented Douchi were more abundant.

In this study, the common and unique flavor substances and metabolites of Douchi fermented by different strains were screened and discussed. This provided a reference for the research of flavor and functional substances of Douchi and the development of new soybean fermented functional food.

## Figures and Tables

**Figure 1 foods-13-03521-f001:**
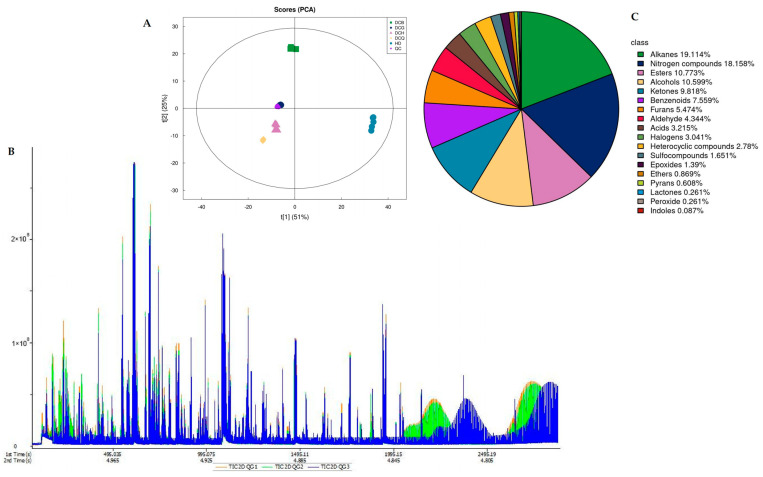
Quality control and sample group analysis of GC×GC–TOFMS. (**A**) PCA analysis of different groups of Douchi by GC×GC–TOFMS; (**B**) total ion current chromatogram of GC×GC–TOFMS; (**C**) classification of identified compounds by GC×GC–TOFMS.

**Figure 2 foods-13-03521-f002:**
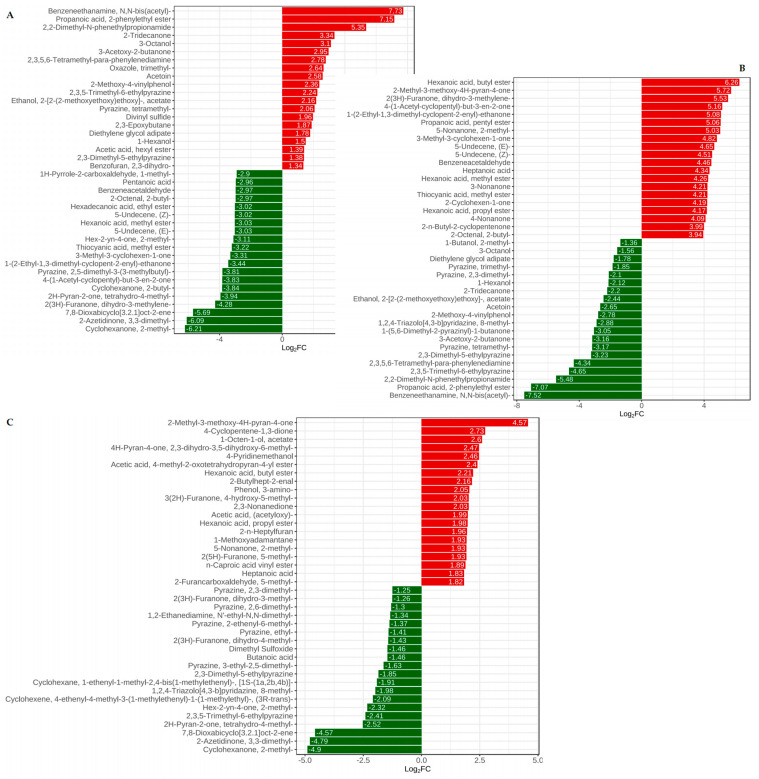
The histogram of main volatile components in Douchi fermented by different strains. (**A**) The histogram of main different volatile components in Douchi fermented by *R. arrhizus* and *B. circulans*; (**B**) the histogram of main different volatile components in Douchi fermented by *A. niger* and *B. circulans*; (**C**) the histogram of main different volatile components in Douchi fermented by *A. niger* and *R. arrhizus*.

**Figure 3 foods-13-03521-f003:**
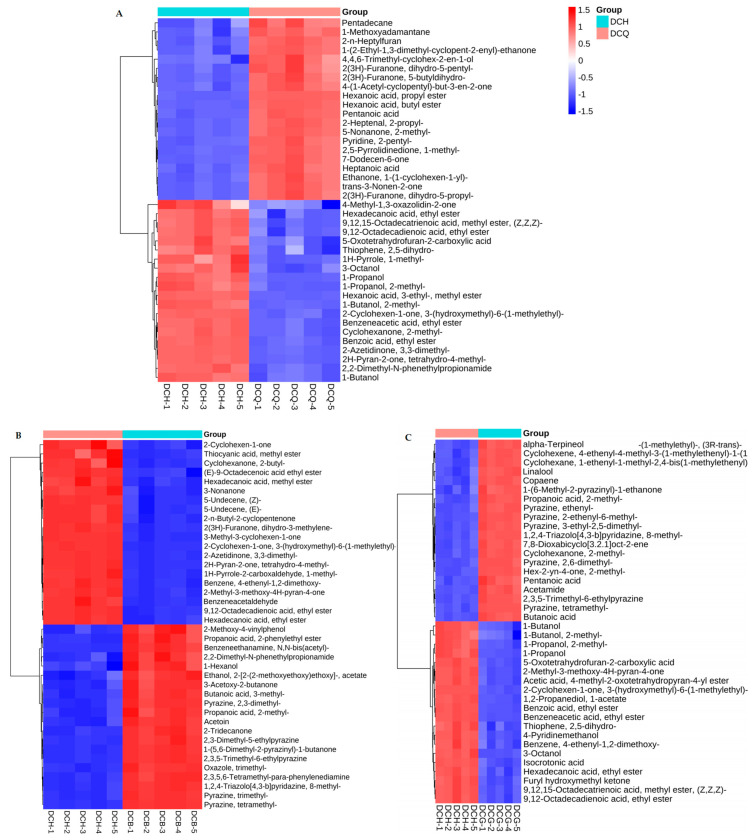
The hierarchical clustering heat map of volatile components with significant differences in Douchi fermented by multi-strain and three pure strains. (**A**) The hierarchical clustering heat map of main different volatile components in Douchi fermented by multi-strain and *A. niger*; (**B**) the hierarchical clustering heat map of main different volatile components in Douchi fermented by multi-strain and *B. circulans*; (**C**) the hierarchical clustering heat map of main different volatile components in Douchi fermented by multi-strain and *R. arrhizus*.

**Figure 4 foods-13-03521-f004:**
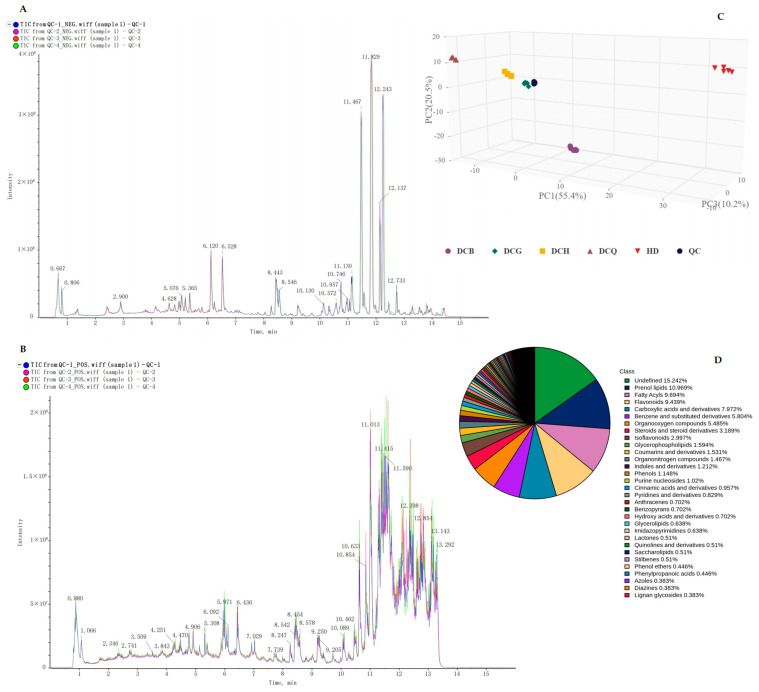
Quality control and sample group analysis of UPLC–Q-TOFMS. (**A**) Total ion current chromatogram of UPLC–Q-TOFMS in negative ion mode; (**B**) total ion current chromatogram of UPLC–Q-TOFMS in positive ion mode; (**C**) PCA analysis of different groups of Douchi by UPLC–Q-TOFMS; (**D**) classification of identified compounds by GC×GC–TOFMS.

**Figure 5 foods-13-03521-f005:**
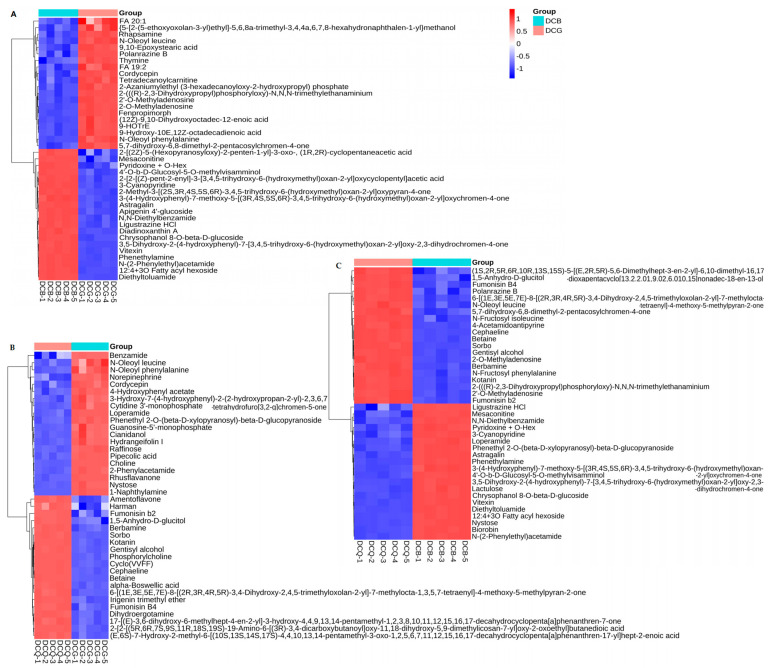
The histogram of significant differences of non-volatile components in Douchi fermented by different strains. (**A**) The differences of main non-volatile components in Douchi fermented by *B. circulans* and *R. arrhizus*; (**B**) the differences of main non-volatile components in Douchi fermented by *A. niger* and *R. arrhizus*; (**C**) the differences of main non-volatile components in Douchi fermented by *A. niger* and *B. circulans*.

**Figure 6 foods-13-03521-f006:**
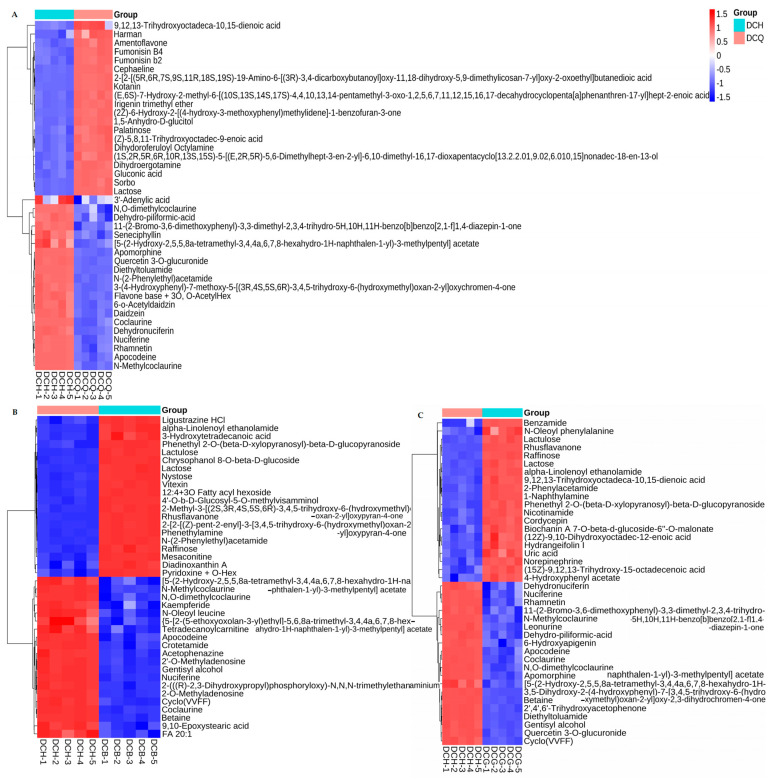
The histogram of significant differences of non-volatile components in Douchi fermented by multi-strain and three pure strains. (**A**) The differences of main non-volatile components in Douchi fermented by multi-strain and *A. niger*; (**B**) the differences of main non-volatile components in Douchi fermented by multi-strain and *B. circulans*; (**C**) the differences of main non-volatile components in Douchi fermented by multi-strain and *R. arrhizus*.

**Figure 7 foods-13-03521-f007:**
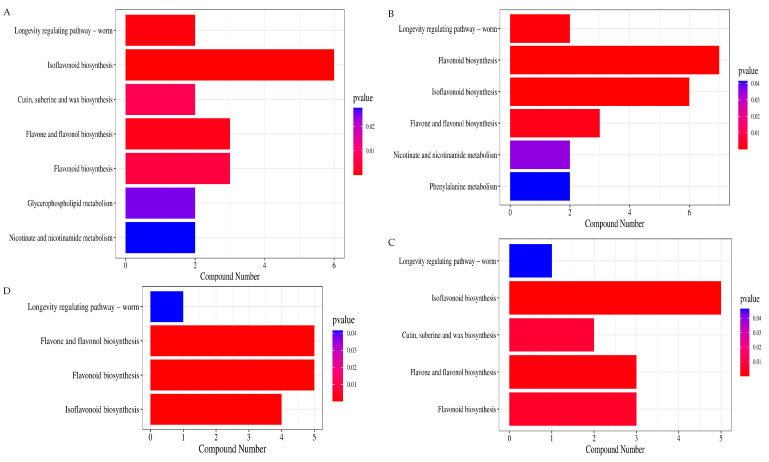
The enriched KEGG pathways of the three strains fermented Douchi and multi-strain fermented Douchi. (**A**) The enriched KEGG pathways of *R. arrhizus* fermented Douchi; (**B**) the enriched KEGG pathways of *B. circulans* fermented Douchi; (**C**) the enriched KEGG pathways of *A. niger* fermented Douchi; (**D**) the enriched KEGG pathways of multi-strain fermented Douch.

## Data Availability

The original contributions presented in the study are included in the article, further inquiries can be directed to the corresponding authors.

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
