# Peer review of "Analysis of Different Strains Fermented Douchi by GC×GC-TOFMS and UPLC–Q-TOFMS Omics Analysis"

_foods, 2024, doi:10.3390/foods13213521_

Round 1
Reviewer 1 Report
Comments and Suggestions for Authors
This MS entitled " Analysis of Different Strains Fermented Douchi by GC×GC-2 TOFMS Omics and Nontarget Metabolomics" covered the phytochemical analysis of Douchi and can draw public attention if following amendment will be followed:
1. Firstly, the study protocol follows GC-MS analysis. GC-MS analysis doesn't cover a wide range of chemicals, especially those with higher molecular weights and non-volatiles. Thus, GC-MS analysis is very preliminary analysis to claim metabolomics accuracy. This study is recommended to run through HPLC analysis against standards to identify compounds. After that, LC-MS/MS or UPLC QTOF MS is recommended to ascertain the molecular weights of single compounds traced in HPLC. This work has UPLS QTOF MS but not to analysis single compound. After HPLC/pHPLC single compound should be analyzed through LC-MS/MS and if the mass value will be similar , you can claim that the compound you identified in HPLC is actually that very compound. As, several compounds may exhibit same retention time though their mass value will be distinctive.
2. Find proper scientific names of plants from this site: http://theplantlist.org/tpl/search
3. Give a brief description of fermentation process, as it is the cruical part of this study, and regulation of this process can yield different results. Thus, proper description is necessary to ensure reproducibility of the study.
4. Plot the chromatograms generated from GC-MS which will shed light on volatile substances.
5. Add future recommendations and limitations of the study.
6. Enrich discussion section and make a bridge of previous reports and this study's outcomes.
7. There are several typos. Please recheck.
Comments on the Quality of English Language
Minor editing of English language required.
Author Response
We sincerely appreciate the time and effort that you have dedicated to providing valuable feedback on our manuscript. We have carefully considered all the comments and have made the necessary revisions to address the concerns raised. Below, we provide a detailed response to each comment.
Comment1: Firstly, the study protocol follows GC-MS analysis. GC-MS analysis doesn't cover a wide range of chemicals, especially those with higher molecular weights and non-volatiles. Thus, GC-MS analysis is very preliminary analysis to claim metabolomics accuracy. This study is recommended to run through HPLC analysis against standards to identify compounds. After that, LC-MS/MS or UPLC QTOF MS is recommended to ascertain the molecular weights of single compounds traced in HPLC. This work has UPLS QTOF MS but not to analysis single compound. After HPLC/pHPLC single compound should be analyzed through LC-MS/MS and if the mass value will be similar, you can claim that the compound you identified in HPLC is actually that very compound. As, several compounds may exhibit same retention time though their mass value will be distinctive.
Response:We thank the reviewer for this insightful comment and agree with the reviewer’s suggestion. Non-target omics analysis can perform unbiased, large-scale and systematic detection of various metabolites in the sample and reflect the changes of metabolites in organisms to the greatest extent, relying on the high-resolution mass analyzer of high-resolution mass spectrometer, which is suitable for the basic research in the early stage of the project. There are also limitations of this method, such as the lack of further identification of most metabolic analysis results, the possibility of partial deletion of metabolites and the limited known metabolic pathways.
This study did not identify compounds against standards screened by non-targeted metabolomics due to the limitation of research funds and time. We have presented this as a deficiency of this study in the discussion section. We will further identify the valuable compounds in subsequent studies. In addition, we have presented the compounds screened by non-target metabolomics in supporting materials, other researchers can find valuable compounds from them.
This revision can be found in the revised manuscript on page 11, line 351-361; supporting information on page 6-9.
Comment2: Find proper scientific names of plants from this site:http://theplantlist.org/tpl/search.
Response:Thank you for pointing this out. We have added proper scientific names of plants from http://theplantlist.org/tpl/search. This revision can be found in the revised manuscript on page 1, line 41.
Comment3: Give a brief description of fermentation process, as it is the cruical part of this study, and regulation of this process can yield different results. Thus, proper description is necessary to ensure reproducibility of the study.
Response:We agree with the reviewer’s suggestion and have added a brief description of fermentation process of Douchi. This revision can be found in the revised manuscript on page 3, line 101-111.
Comment4: Plot the chromatograms generated from GC-MS which will shed light on volatile substances.
Response: We agree with the reviewer’s suggestion and have ploted the chromatograms generated from GC-MS of Douchi fermented by different strains. This revision can be found in supporting information on page 2.
Comment5: Add future recommendations and limitations of the study.
Response: We appreciate the reviewer’s feedback on this matter. We have added future recommendations and limitations of the study. This revision can be found in the revised manuscript on page11, line 358-361.
Comment6: Enrich discussion section and make a bridge of previous reports and this study's outcomes.
Response: We appreciate the reviewer’s feedback on this matter. We have enriched discussion section and made a bridge of previous reports and this study's outcomes. This revision can be found in the revised manuscript on page 11, line 343-350.
Comment7: There are several typos. Please recheck.
Response: Thank you for pointing this out. We have rechecked and modified the typos.This revision can be found in the revised manuscript on page 1, line 33,34, 43; page 2, line 66, 69, 70, 72, 87; page 6, line 226-230; page 7, line 234-257; page 8, line 268, 271-273; page 9, line 290; page 10, line 320-327; page 11, line 329-340; page 12, line 385-387.
We have also made several other improvements throughout the manuscript to enhance clarity and readability. These include the numbers of co-corresponding authors according to the processing regulation of Foods.However, the order and contributions of authors have not changed.
We hope that the revised manuscript meets your expectations. We believe that the changes have significantly improved the quality of our work. Thank you once again for your time and consideration.
Reviewer 2 Report
Comments and Suggestions for Authors
The manuscript "Analysis of Different Strains Fermented Douchi by GC×GC-TOFMS Omics and Nontarget Metabolomics" is dealing with fermentation of black soybean with different strains Aspergillus niger, Rhizopus arrhizus and Bacillus circulans. Final goal is provided a reference for the research of flavor and 36 functional substances of Douchi. However, all the figures are very small, indistinct. I suggest that the chemical analysis be shown in tabular form as usual.
Additionally, I suggest using CAS numbers because some of the compounds have very long and complicated names (such as (3S,8As)-3-isobutylhexahydropyrrolo[1,2-a]pyrazine-1,4-dione; 2,3-dihydro -3,5-dihydroxy-6-methyl-23 4H-pyran-4-one; etc).
Author Response
We sincerely appreciate the time and effort that you have dedicated to providing valuable feedback on our manuscript. We have carefully considered all the comments and have made the necessary revisions to address the concerns raised. Below, we provide a detailed response to each comment.
Comment1: However, all the figures are very small, indistinct. I suggest that the chemical analysis be shown in tabular form as usual.
Response:Thank you for pointing this out. We have shown chemical analysis in tabular form as usual. This revision can be found in supporting information on page 2-9.
Comment2: Additionally, I suggest using CAS numbers because some of the compounds have very long and complicated names such as (3S,8As)-3-isobutylhexahydropyrrolo [1,2-a]pyrazine-1,4-dione, 2,3-dihydro -3,5-dihydroxy-6-methyl-23 4H-pyran-4-one, etc.
Response:We appreciate the reviewer’s feedback on this matter. To address this, we have searched CAS numbers of the compounds with very long and complicated names, and used CAS number instead of compound name. This is reflected in the revised manuscript on page 1, line 20,23; page 5, line 197, 198; page 7, line 252; page 8, line 279; page 11, line 370,373; page 12, line 384.
We have also made several other improvements throughout the manuscript to enhance clarity and readability. These include the numbers of co-corresponding authors according to the processing regulation of Foods.However, the order and contributions of authors have not changed.
We hope that the revised manuscript meets your expectations. We believe that the changes have significantly improved the quality of our work. Thank you once again for your time and consideration.
Round 2
Reviewer 1 Report
Comments and Suggestions for Authors
Can be accepted in current form
Comments on the Quality of English LanguageMinor english editing is required
Author Response
Comment 1: The figures contain too much information which make them not clear and its fonts are too small. Thus, I urge the Authors to reconsider the way they present their data.
Response:We thank the reviewer for this insightful comment and agree with the reviewer’s suggestion. We have re-edited the figures and displayed the top 20 compounds of volatile components and non-volatile components in different strains fermented Douchi for clearer figures. This revision can be found in the revised manuscript on page 4-11.
Comment 2: Appeals should be equal for both analytical techniques (GC×GC-TOFMS, UPLC–Q-TOFMS) in the title, abstract, introduction, and so on. Reconsider the use of the term "non-target metabolomics analysis". The GC×GC-TOFMS analysis could also be a non-target approach. I suggest to differentiate the two analyses by their chromatography natures or by their target compounds (e.g., volatiles and non-volatile metabolites, respectively).
Response: Thank you for pointing this out. We have expressed "non-target metabolomics analysis" by "UPLC–Q-TOFMS omics analysis". The methods have been divided into volatile components analysis and non-volatile components analysis.
This revision can be found in the revised manuscript on page 1, line 3, 16, 37; page 3, line 129; page 12, line 302.
Comment 3: I think the writing of full name (other name) of a compound is necessary, and therefore I suggest the Authors to rewrite the names of compound in their manuscript, not as a CAS number.
Response:We agree with the reviewer’s suggestion and have added full name of the compound. This revision can be found in the revised manuscript on page 1, line 20,22; page 5, line 179, 180; page 8, line 226; page 10, line 263; page 12, line 346,351; page 13, line 362.
Comment 4: Scientific name of microorganism/plant should be written full in the first mention, and then the genus should be abbreviated after the second mention. For example, first mention as Aspergillus niger; after second mention as A. niger..
Response: We agree with the reviewer’s suggestion and have modified this problem according to the reviewer. This revision can be found in supporting information in the revised manuscript on page 1-13.
Comment 5: Same comment for instrument names, and so on. For example, first mention gas chromatography (GC); after second mention as GC.
Response: We appreciate the reviewer’s feedback on this matter. have modified this problem according to the reviewer. This revision can be found in the revised manuscript on page2, line 73, 78, 86, 87; page3, line 116, 133.
Comment6: Page 1 Line 41: "Tianpei" should be "Tempeh".
Response: We appreciate the reviewer’s feedback on this matter. We have modified the problem according to the reviewer. This revision can be found in the revised manuscript on page 1, line 42.
Comment7: 2.3. Volatile Component Analysis: What is the standard compound(s) for quantification? Mention the version of NIST library!
Response: Thank you for pointing this out. The quantification was relative percentage content based on the peak areas, we have modified the expression of this problem. We have rechecked and added the version of NIST library. This revision can be found in the revised manuscript on page 3, line 127.
Comment8: 2.4. Non-Target Metabolomics Analysis: What is MSDAIL? Is it MS-DIAL? Mention its version and library(s) used for data analysis.
Response:Thank you for pointing this out. We have rechecked and modified the problem. This revision can be found in the revised manuscript on page 5, line 147, 148, 150.